# Cytogenetic and Molecular Marker Analyses of a Novel Wheat–*Psathyrostachys huashanica* 7Ns Disomic Addition Line with Powdery Mildew Resistance

**DOI:** 10.3390/ijms231810285

**Published:** 2022-09-07

**Authors:** Binwen Tan, Miaomiao Wang, Li Cai, Sanyue Li, Wei Zhu, Lili Xu, Yi Wang, Jian Zeng, Xing Fan, Lina Sha, Dandan Wu, Yiran Cheng, Haiqin Zhang, Guoyue Chen, Yonghong Zhou, Houyang Kang

**Affiliations:** 1State Key Laboratory of Crop Gene Exploration and Utilization in Southwest China, Sichuan Agricultural University, Chengdu 611130, China; 2Triticeae Research Institute, Sichuan Agricultural University, Chengdu 611130, China; 3College of Resources, Sichuan Agricultural University, Chengdu 611130, China; 4College of Grassland Science and Technology, Sichuan Agricultural University, Chengdu 611130, China

**Keywords:** *Psathyrostachys huashanica*, alien addition line, powdery mildew, GBS, PCR-based marker

## Abstract

Powdery mildew caused by *Blumeria graminis* f. sp. *tritici* is a devastating disease that reduces wheat yield and quality worldwide. The exploration and utilization of new resistance genes from wild wheat relatives is the most effective strategy against this disease. *Psathyrostachys huashanica* Keng f. ex P. C. Kuo (2*n* = 2*x* = 14, NsNs) is an important tertiary gene donor with multiple valuable traits for wheat genetic improvement, especially disease resistance. In this study, we developed and identified a new wheat—*P. huashanica* disomic addition line, 18-1-5—derived from a cross between *P. huashanica* and common wheat lines Chinese Spring and CS*ph2b*. Sequential genomic and multicolor fluorescence in situ hybridization analyses revealed that 18-1-5 harbored 21 pairs of wheat chromosomes plus a pair of alien Ns chromosomes. Non-denaturing fluorescence in situ hybridization and molecular marker analyses further demonstrated that the alien chromosomes were derived from chromosome 7Ns of *P. huashanica*. The assessment of powdery mildew response revealed that line 18-1-5 was highly resistant at the adult stage to powdery mildew pathogens prevalent in China. The evaluation of agronomic traits indicated that 18-1-5 had a significantly reduced plant height and an increased kernel length compared with its wheat parents. Using genotyping-by-sequencing technology, we developed 118 PCR-based markers specifically for chromosome 7Ns of *P. huashanica* and found that 26 of these markers could be used to distinguish the genomes of *P. huashanica* and other wheat-related species. Line 18-1-5 can therefore serve as a promising bridging parent for wheat disease resistance breeding. These markers should be conducive for the rapid, precise detection of *P. huashanica* chromosomes and chromosomal segments carrying *Pm* resistance gene(s) during marker-assisted breeding and for the investigation of genetic differences and phylogenetic relationships among diverse Ns genomes and other closely related ones.

## 1. Introduction

Powdery mildew, caused by the biotrophic fungal pathogen *Blumeria graminis* f. sp. *tritici* (*Bgt*), is a devastating disease that poses a serious threat to wheat production worldwide [1]. The pathogen interferes with photosynthesis by infecting wheat leaves, sheaths, stems, and ears and affects plant growth, heading, and grain filling, thereby giving rise to significant yield losses—13–34% in years of average infection and up to 50% in years of severe epidemics [2,3]. Host resistance is considered to be the most effective, economical, and environmentally friendly measure to control losses caused by the disease [4]. Thus far, 68 formally designated wheat powdery mildew resistance (*Pm1–Pm68*) genes and more than 100 quantitative trait loci (QTLs) have been documented [5,6,7]. Forty-seven of these genes originated from progenitors and wild relatives of wheat, including *Triticum monococcum*, *T. boeoticum*, *T. carthlicum*, *T. turgidum*, *T. urartu*, *T. timopheevii*, *Aegilops speltoides*, *Ae. tauschii*, *Ae. longissima*, *Ae. searsii*, *Ae. ovata*, *Secale cereale*, *Dasypyrum villosum*, *Thinopyrum intermedium*, *Th. ponticum*, and *Agropyron cristatum* [7,8,9,10]. Some of these resistance genes, such as *Pm12* from *Ae. speltoides* and *Pm21* from *D. villosum*, which both confer broad-spectrum resistance to many *Bgt* isolates, have played important roles in wheat production [11,12]. Because of the coevolution of pathogen virulence with host resistance and the rapid emergence of new virulent isolates, however, most powdery mildew resistance genes deployed in large-scale commercial production have successively lost their effectiveness against the pathogen [4,13]. A typical example is the “boom–bust” of *Pm8* derived from chromosome arm 1RS of Petkus rye, which has triggered continual severe epidemics of powdery mildew in the main wheat production regions of China [14,15]. The discovery, identification, and deployment of additional sources of powdery mildew resistance in wheat-related germplasm to address this vulnerability is therefore urgently needed for breeding new resistant wheat cultivars [16].

*Psathyrostachys huashanica* Keng f. ex P. C. Kuo (2*n* = 2*x* = 14, NsNs), an important wild relative of wheat, is restricted to the stony slopes of the Huashan Pass in the Qingling Mountains of Shaanxi Province, China [17,18]. This endemic species has attracted considerable attention from wheat breeders because it possesses numerous excellent characteristics, such as resistance to multiple wheat diseases (scab, stripe rust, powdery mildew, and take-all), tolerance to abiotic stress (drought, salinity, and alkalinity), early maturation, and dwarf stature [19,20,21], which make it a potentially valuable genetic reservoir for wheat improvement. With the aim of making these favorable genes of *P. huashanica* accessible for wheat breeding, intergeneric hybrids between common wheat and *P. huashanica* were successfully synthesized through distant hybridization and embryo rescue in the 1980s [19]. Subsequently, an array of wheat–*P. huashanica* derivative lines were successively generated, including intergeneric amphiploids [22], addition [23,24,25], substitution [26,27], and translocation [20,28,29] lines. Compared with their recipient wheat parents, these progeny lines with single *P. huashanica* chromosomes or chromosomal segments incorporated into the common wheat background have performed extremely well and have thus proven to be valuable germplasm resources for wheat genetic improvement. Despite these successes, the usefulness of powdery mildew resistance genes from *P. huashanica* in wheat breeding programs has been underexplored to date.

In previous research in our laboratory, we successfully introduced desirable genes from *P. huashanica* into common wheat by intergeneric hybridization to produce a series of wheat–*P. huashanica* derived lines, including addition, translocation, and some unrecognized hybrid lines [20,25,30]. In the present study, we selected 18-1-5, a novel wheat–*P. huashanica* disomic addition line highly resistant to powdery mildew, from the BC_1_F_5_ progeny of a cross between *P. huashanica* and common wheat lines Chinese Spring (CS) and a CS *ph2b* mutant (CS*ph2b*). The main objectives of our study were to: (1) characterize the chromosomal composition of 18-1-5 by in situ hybridization and molecular marker analyses, (2) evaluate the powdery mildew resistance and agronomic performance of 18-1-5, and (3) develop and validate new PCR-based molecular markers specific for the alien chromosome of 18-1-5 using genotyping-by-sequencing (GBS) technology. These markers should be useful for efficiently tracing *P. huashanica* 7Ns chromosomes and chromosomal segments during marker-assisted selection to improve wheat disease resistance.

## 2. Results

### 2.1. Chromosomal Composition of 18-1-5

Sequential genomic in situ hybridization (GISH) and multicolor fluorescence in situ hybridization (mc-FISH) analyses were performed to determine the chromosomal constitution of wheat–*P. huashanica* line 18-1-5. GISH analysis using whole genomic DNA of *P. huashanica* as a probe and CS as a blocker DNA demonstrated that line 18-1-5 had 44 chromosomes: 42 emitting only the blue DAPI signals and a pair of intact chromosomes with bright-red hybridization signals (Figure 1a), the latter indicating that line 18-1-5 contained two chromosomes from *P. huashanica*. Following GISH analysis, mc-FISH with two probes, Oligo-pSc119.2 (green) and Oligo-pTa535 (red), was carried out to characterize the wheat chromosomes in line 18-1-5. By comparison with the standard FISH karyotype of CS [31], all 21 wheat chromosome pairs were successfully identified (Figure 1b). When pPh37 (green) was used as a probe for the FISH analysis, two strong terminal signals were observed on the short and long arms of the additional *P. huashanica* chromosomes, but no hybridization signals were detected on the 42 wheat chromosomes (Figure 1c,d). These results suggested that 18-1-5 was a wheat–*P. huashanica* disomic addition line.

To further confirm the cytological stability of line 18-1-5, we used FISH and GISH to characterize 50 randomly selected seeds representing selfed progeny of 18-1-5. Forty-six seeds were found to carry a pair of Ns chromosomes (Figure 1e,f), whereas four seeds harbored one Ns chromosome (Figure 1g,h). This result indicated that 18-1-5 was a cytogenetically relatively stable wheat–*P. huashanica* disomic addition line.

### 2.2. Non-Denaturing Fluorescence In Situ Hybridization (ND-FISH) Analysis of 18-1-5

ND-FISH analysis with probes Oligo-pSc200, Oligo-44, and Oligo-pTa71A-2 was performed on line 18-1-5. A pair of chromosomes produced strong terminal pSc200 signals on both arms and distinct Oligo-44 signals in the middle region of both arms (Figure 2a). According to the published FISH karyotype of *P. huashanica* [32], this pair of chromosomes in line 18-1-5 were identified as 7Ns chromosomes. At the same time, sequential mc-FISH and GISH analyses of the same metaphase cell also demonstrated that the two above-mentioned chromosomes were Ns chromosomes from *P. huashanica* (Figure 2b,c). These findings therefore suggested that 18-1-5 was a wheat–*P. huashanica* 7Ns disomic addition line with a chromosomal composition of 2*n* = 44 = 21″ W + 1″ Ns (7Ns).

### 2.3. Molecular Marker Analysis of 18-1-5

We tested 135 pairs of PLUG markers and found that six markers (*TNAC1782*, *TNAC1803*, *TNAC1815*, *TNAC1867*, *TNAC1806*, and *TNAC1845*) distributed on the seventh homoeologous group of chromosomes of wheat were able to amplify apparently identical polymorphic regions in 18-1-5 and *P. huashanica* but not in the wheat parents CS and CS*ph2b* (Figure 3a–f, Appendix A). Six SLAF-based markers (i.e., PH7Ns-9, PH7Ns-14, PH7Ns-38, PH7Ns-43, PH7Ns-48, and PH7Ns-49) specific for *P. huashanica* chromosome 7Ns also amplified the same specific regions from 18-1-5 and *P. huashanica* but not from CS and CS*ph2b* (Figure 3g–l, Appendix A). These results clearly demonstrated once again that the alien chromosomes in line 18-1-5 were the 7Ns chromosomes of *P. huashanica*.

### 2.4. Powdery Mildew Response of 18-1-5

The powdery mildew responses of 18-1-5, CS, CS*ph2b*, *P. huashanica*, MY11, and L658 to a mixture of *Bgt* isolates were assessed at the adult stage in a controlled-environment room. Our analysis revealed that 18-1-5 was highly resistant (IT = 1), whereas *P. huashanica* and the resistant control L658 were immune (IT = 0). In contrast, the wheat parents CS and CS*ph2b* were highly susceptible (IT = 8); the same was true for the susceptible check MY11 (IT = 9) (Figure 4).

### 2.5. Agronomic Performance of 18-1-5

The agronomic traits of 18-1-5 and its wheat parents were investigated in an experimental field. The line 18-1-5 displayed stable morphological traits, which were similar to those of the wheat parents CS and CS*ph2b* (Figure 5, Table 1). The average plant height of 18-1-5 was significantly smaller than that of CS and CS*ph2b*, whereas the average kernel length was significantly larger. Furthermore, spikelets per spike, kernels per spike, kernel width, and thousand-kernel weight were significantly lower in 18-1-5 than in CS and CS*ph2b*. No significant differences in tiller number or spike length were observed between 18-1-5 and either CS or CS*ph2b*.

### 2.6. Acquisition of Specific P. huashanica 7Ns Chromosome Sequences

A total of 1,404,766,080, 1,809,454,176, and 2,734,574,112 raw reads based on GBS technology were obtained from CS*ph2b*, *P. huashanica* ZY3156, and 18-1-5, respectively. The details of the sequencing data are summarized in Table 2. Average Q20 and Q30 scores were 95.14% and 87.53%, respectively, and the average GC content was 44.14%. After filtering out low-depth data, 9,754,818, 12,565,464, and 18,988,190 effective GBS reads, with an average sequencing depth of 20.59×, were finally acquired for CS*ph2b*, *P. huashanica* ZY3156, and 18-1-5, respectively (Table 3). Using the Burrows–Wheeler Alignment tool, we determined that 429,274 reads from 18-1-5 had less than 18% similarity to the CS reference genome. Among these reads, 15,925 shared less than 18% similarity with CS*ph2b* but at least 90% similarity with *P. huashanica* and were therefore considered to be specific segments of chromosome 7Ns from *P. huashanica*.

### 2.7. Development and Validation of Specific Molecular Markers

To develop *P. huashanica* 7Ns chromosome-specific markers, we designed primers based on the 573 fragments most similar to *P. huashanica* sequences (Appendix A). In total, 118 primer pairs, including primers for markers GPH7Ns-155, GPH7Ns-300, and GPH7Ns-523, amplified specific regions in 18-1-5 and *P. huashanica* but not in CS and CS*ph2b* (Figure 6a–c). These markers, representing 20.4% of developed primers, were therefore regarded as specific for *P. huashanica* chromosome 7Ns, the alien chromosome.

To evaluate the specificity and stability of the 118 chromosome 7Ns-specific markers, we used these markers in a subsequent PCR analysis of 13 wheat-related species (Appendix A). Twenty-six of these markers yielded specific amplification bands when applied to *P. huashanica* but not when applied to other related analyzed species (Figure 7a) and could thus be used to distinguish *P. huashanica* from other wheat-related species. In contrast, three markers amplified specific regions not only in *P. huashanica* but also in *P. juncea* (Figure 7b). In addition, 42 markers amplified the same specific fragments in *P. huashanica*, *P. juncea*, *L. multicaulis*, and *L. angustus* but not in any other wheat-related species (Figure 7c). PCR amplification bands were also obtained from other wheat-related species using some of the other markers. In particular, 3, 2, 7, 8, 14, 7, 12, 6, and 11 markers amplified specific regions of *T. monococcum*, *Ae. tauschii*, *Th. bessarabicum*, *Th. elongatum*, *D. villosum*, *H. bogdanii*, *Ag. cristatum*, *S. cereale*, and *Pse. libanotica*, respectively. For example, the marker GPH7Ns-350 amplified specific regions of *Ag. cristatum* (Figure 7d).

## 3. Discussion

In wheat breeding programs, distant hybridization is an important approach for transferring genes that enhance wheat productivity from wild relatives into common wheat varieties and for broadening the genetic base of wheat cultivars [33]. Wheat-alien chromosome addition lines are usually used as important bridge materials for the transfer of elite alien genes [34]. *Psathyrostachys huashanica*, which represents a tertiary gene pool for wheat containing numerous beneficial genes for improving the tolerance of common wheat to biotic and abiotic stresses, has been used extensively for distant hybridization with common wheat. Previous studies have illustrated that all seven Ns chromosomes from *P. huashanica* that have been introduced into common wheat exhibit excellent characteristics with regard to disease resistance and agronomic traits. Examples include 2Ns, 3Ns, 4Ns, and 5Ns disomic addition lines and a 1Ns (1D) disomic substitution line with resistance to stripe rust [35,36,37,38], 1Ns and 7Ns disomic addition lines with resistance to leaf rust [24,39], 6Ns and 7Ns addition lines maturing earlier than their wheat parents [25,40], and a 2Ns (2D) disomic substitution line with a high level of resistance to wheat take-all [41]. In addition, a 4Ns disomic addition line has been found to have enhanced tiller numbers [38], and a 1Ns disomic addition line was reported to exhibit increased storage of microelements in seeds [42]. Li et al. [28] recently developed a cytogenetically stable wheat–*P. huashanica* T3DS-5NsL·5NsS and T5DL-3DS·3DL dual translocation line with much better powdery mildew resistance than its wheat parents at both seedling and adult stages. Han et al. [43] identified a wheat–*P. huashanica* 1Ns disomic addition line, H5-5-4-2, that was highly resistant to powdery mildew at both seedling and adult stages. Liu et al. [29] generated a new wheat–*P. huashanica* T3DS·3DL-4NsL and T3DL·4NsS translocation line exhibiting a high level of resistance to powdery mildew at the adult stage and concluded that the resistance was derived from chromosome 4Ns. DH109, a novel wheat–*P. huashanica* 3Ns (3D) disomic substitution line with superior resistance to powdery mildew at the seeding stage, was identified from the F_7_ progeny of wheat–*P. huashanica* heptaploid line H8911 × durum wheat Trs-372 [26]. Unfortunately, only a few wheat–*P. huashanica*-derived lines with resistance to powdery mildew are available for wheat breeding. In the current study, we used GISH, mc-FISH, ND-FISH, and molecular marker analyses to successfully develop and identify a novel wheat–*P. huashanica* 7Ns disomic addition line, 18-1-5, derived from the BC_1_F_5_ progeny of *P. huashanica*/CS*ph2b*//CS. The evaluation of powdery mildew resistance suggested that 18-1-5 was highly resistant at the adult stage to a mixture of *Bgt* isolates. In addition, *P. huashanica* was immune, whereas the wheat parents CS and CS*ph2b* were susceptible. These results indicate that the powdery mildew resistance of 18-1-5 is derived from *P. huashanica*, according to the only evidence provided by this pedigree.

Years of intense effort and numerous pieces of evidence, including observations of meiotic pairing in interspecific hybrids, DNA hybridization patterns, DNA sequence information, and phylogenetic analysis, have confirmed that *Leymus* Hochst. shares the same Ns genome from *Psathyrostachys* Nevski [44,45,46]. Some wheat–*Leymus mollis* derived lines encompassing alien Ns chromosomes have also been produced, such as two 3Ns (3D) disomic substitution lines with resistance to either leaf rust or powdery mildew [26,47], a 1Ns (1D) disomic substitution line with improved protein and glutenin contents [48], a 2Ns, 3Ns (2D, 3D) double substitution line with resistance to yellow rust and Fusarium head blight [49], and two 2Ns (2D) and 7Ns (7D) disomic substitution lines resistant to stripe rust [50,51]. Nonetheless, no relevant reports describing the transfer of powdery mildew resistance genes from chromosome 7Ns of *Leymus* or *Psathyrostachys* to common wheat have appeared. In the present study, we accomplished the successful transfer of a high-level powdery mildew resistance gene from chromosome 7Ns of *P. huashanica*. Furthermore, our evaluation of agronomic traits suggested that 18-1-5 has a significantly reduced plant height and increased kernel length compared with its wheat parents. In view of the high variability of pathogens and the uniformity of resistance sources, which have resulted in the rapid loss of the vast majority of *Pm* genes, the disomic addition line 18-1-5, highly resistant to powdery mildew and possessing desirable agronomic traits, is therefore a novel, promising germplasm resource for the genetic improvement of wheat powdery mildew resistance. 

Molecular markers have been extensively applied to quickly and accurately trace alien chromosomes or chromosomal segments in a wheat background, in turn greatly improving the selection efficiency of marker-assisted breeding and shortening the breeding cycle [52]. An increasing number of potentially beneficial genes have been gradually identified in *P. huashanica*, but the lack of specific, stable, efficient, and reliable molecular markers imposes significant restrictions on their further utilization in breeding [53]. The development of molecular markers specific for *P. huashanica* is thus important and urgently needed to facilitate future research efforts. *Psathyrostachys huashanica* genome- or chromosome-specific molecular markers previously developed by conventional methodologies, including random-amplified polymorphic DNA and sequence-characterized amplified region techniques, are inefficient, inaccurate, and costly [21,24,54,55,56,57,58]. With the rapid advancement of next-generation sequencing technologies and the declining cost of genome sequencing, tackling these weaknesses has fortunately become possible. For instance, 25 specific PCR markers for *D. villosum* chromosome 6V#4S carrying the *PmV* gene have been developed by RNA-seq [59]. As another example, we previously used SLAF-seq technology to develop 45 markers specific for one arm of *P. huashanica* chromosome 7Ns to efficiently identify 7Ns chromosome segments carrying earliness per se genes in a wheat background [25]. Compared with the above-mentioned technologies, GBS is a simpler, more robust, and more cost-effective approach [60]; it is also more suitable for the development of large-scale molecular markers in Triticeae species without reference genome sequences, such as for tetraploid *Thinopyrum elongatum* chromosomes 1E and 4E [53,61]. The use of GBS for the development of *P. huashanica*-specific molecular markers has not been reported until now, and the number of existing markers has also been totally insufficient to satisfy research demands. In the present study, we produced 118 PCR-based markers specific for *P. huashanica* chromosome 7Ns in line 18-1-5 using a GBS approach. These markers may be applied to trace chromosome 7Ns carrying the powdery mildew resistance gene(s) and can also augment available markers for *P. huashanica*. In addition, we obtained 26 markers that can be used to clearly distinguish the genomes of *P. huashanica* and other wheat-related species. Only three of these markers amplified a common sequence in *P. huashanica* and *P. juncea*, which indicates that the Ns genomes of these two species have greatly diverged even though they belong to the same genus. This result is consistent with previous findings based on C-banding and cytogenetic analyses reported by Lu et al. [62] and Wang et al. [63]. The most important point is that 42 markers amplified the same specific regions in Ns-genome-containing species. Our findings, which provide further evidence that the genera *Leymus* and *Psathyrostachys* are closely related and that the Ns genome of *Leymus* originated from *Psathyrostachys*, are in accordance with previous inferences [46,64]. Interestingly, marker validation analyses suggested that marker amplification frequencies were considerably higher in Ns and NsXm genomic species than in other wheat-related species involving A, D, E^b^, E^e^, V, H, P, R, and St genomes, which reflects the fact that the Ns genome and other genomes are genetically distant from one another.

## 4. Materials and Methods

### 4.1. Plant Materials

*Psathyrostachys huashanica* accession ZY3156 (2*n* = 2*x* = 14, NsNs) was collected from the Huashan Mountains, Shaanxi Province, China, by C. Yen and J. L. Yang of Sichuan Agricultural University. Two wheat lines were used in this study: the wheat landrace CS (*Triticum aestivum* L., 2*n* = 6*x* = 42, AABBDD), a member of the Sichuan white wheat complex group, and CS*ph2b*, a chemically induced mutant originally produced by Sears [65]. To use the desirable traits of *P. huashanica*, we first crossed *P. huashanica* with CS*ph2b*, and the resulting F_1_ plants were further crossed with CS (as the recurrent parent) to generate a BC_1_F_1_ population in 2005 [30]. Seeds selected from the BC_1_F_1_ plants were then bulked and advanced to the BC_1_F_5_ generation by single seed descent. Ultimately, a wheat–*P. huashanica* 7Ns disomic addition line conferring resistance to powdery mildew, named 18-1-5, was isolated from the BC_1_F_5_ generation. The wheat line Mianyang 11 (MY11) was used as a susceptible check for the assessment of powdery mildew resistance, while the wheat line L658 was used as a resistant control. To verify the specificity and stability of newly developed molecular markers, we used 13 wheat-related species involving diverse genomes: *Triticum monococcum* accession PI168805 (2*n* = 2*x* = 14, AA), *Aegilops speltoides* accession PI542251 (2*n* = 2*x* = 14, SS), *Aegilops tauschii* accession AS84 (2*n* = 2*x* = 14, DD), *Psathyrostachys juncea* accession PI314082 (2*n* = 2*x* = 14, NsNs), *Thinopyrum elongatum* accession PI531718 (2*n* = 2*x* = 14, E^e^E^e^), *Thinopyrum bessarabicum* accession W6-10232 (2*n* = 2*x* = 14, E^b^E^b^), *Dasypyrum villosum* accession PI251477 (2*n* = 2*x* = 14, VV), *Hordeum bogdanii* accession Y1819 (2*n* = 2*x* = 14, HH), *Agropyron cristatum* accession PI610892 (2*n* = 2*x* = 14, PP), *Secale cereale* accession QL (2*n* = 2*x* = 14, RR), *Pseudoroegneria libanotica* accession PI228391 (2*n* = 2*x* = 14, StSt), *Leymus multicaulis* accession PI440325 (2*n* = 4*x* = 28, NsNsXmXm), and *Leymus angustus* accession PI440308 (2*n* = 8*x* = 56, NsNsNsNsXmXmXmXm). Voucher specimens have been deposited in the herbarium of the Triticeae Research Institute, Sichuan Agricultural University, China (SAUTI).

### 4.2. Sequential GISH and Mc-FISH Analyses

Root tips from seeds germinated in Petri dishes on moistened filter paper at 22 °C were excised and treated with nitrous oxide gas for 2 h 30 min and then fixed in glacial acetic acid for 5–10 min. The meristems were then cut and digested with pectinase and cellulase (Yakult Pharmaceutical Ind. Co., Tokyo, Japan) [66]. Slides were prepared for GISH and mc-FISH analyses using a previously described method [67]. Total genomic DNAs were extracted from fresh leaves of *P. huashanica* and CS using the improved cetyltrimethylammonium bromide method [68]. To serve as a probe, *P. huashanica* genomic DNA was labeled by the nick translation method using an Atto550 NT labeling kit (Jena Bioscience, Jena, Germany), whereas CS genomic DNA was used as blocker. The probe to blocker ratio was 1:150. The GISH procedure was conducted according to a published method [69] with minor modifications. Specifically, 20 µL of hybridization solution comprising 16 µL hybridization mixture (20× saline sodium citrate [SSC], salmon sperm DNA, deionized formamide, and dextran hydrogen sulfate sodium salt), 1 μL labeled probe DNA (100 ng/μL), and 3 μL blocking DNA (4000 ng/μL) was loaded on each slide. The slides were denatured at 85 °C for 5 min, incubated at 50 °C for 8 h, washed sequentially in 2× SSC at 55 °C for 10 min and 75% and 100% (*v*/*v*) ethanol for 2 min, and then dried at room temperature. Chromosomes were counterstained with 4,6-diamino-2-phenylindole solution (DAPI; Vector Laboratories, Burlingame, CA, USA), and fluorescence signals were detected and visualized using a fluorescence microscope (Olympus BX63) equipped with a Photometric SenSys DP-70 CCD camera (Olympus, Tokyo, Japan). The photomicrographs were processed using Adobe Photoshop.

After the GISH analysis, the photographed slides were washed sequentially with 75% and 100% (*v*/*v*) ethanol for 5 min, 2× SSC at 60 °C for 30 min, and 75 and 100% (*v*/*v*) ethanol for 5 min, and then exposed to bright light for 48 h to remove GISH signals. For the mc-FISH analysis, the synthetic oligonucleotide probes Oligo-pSc119.2 and Oligo-pTa535 were used to identify individual wheat chromosome [31]. The repetitive DNA probe pPh37, which was labeled by the nick translation method using an Atto488 NT labeling kit, allowed all *P. huashanica* chromosomes to be unambiguously distinguished from wheat chromosomes [25]. The mc-FISH procedure was conducted as described by Li et al. [61] and Han et al. [67] with slight modifications. Hybridization signals were observed and captured as described above. 

### 4.3. ND-FISH Analysis

The probe combination Oligo-pSc200 [70], Oligo-44 [71], and Oligo-pTa71A-2 [72], which was able to clearly distinguish all *P. huashanica* chromosomes from each other and all wheat chromosomes [32], was used to confirm the homoeologous group relationships of the added alien chromosomes in line 18-1-5. These oligonucleotide probes were 5′-end-labeled with 6-carboxytetramethylrhodamine or 6-carboxyfluorescein by Sangon Biotech (Chengdu, China). The ND-FISH analysis, which was followed by sequential mc-FISH and GISH analyses as described above, was performed according to the method described by Fu et al. [70].

### 4.4. Molecular Marker Analysis

A total of 135 pairs of PCR-based landmark unique gene (PLUG) primers distributed evenly on the seven homoeologous groups of wheat [73] were used to determine homoeologous group relationships of the introduced alien chromosomes in line 18-1-5. Six SLAF (specific-locus amplified fragment sequencing)-based markers specific for *P. huashanica* chromosome 7Ns [25] were used to identify the alien chromatin in line 18-1-5. All PCR primers were synthesized by Sangon Biotech. CS and CS*ph2b* were used as negative controls, and *P. huashanica* served as a positive control. PCR amplification and product separation were performed according to previously published methods [25,74]. 

### 4.5. Powdery Mildew Response

The powdery mildew responses of CS, CS*ph2b*, *P. huashanica*, MY11, L658, and 18-1-5 were evaluated at the adult plant stage with three replicates in a controlled greenhouse. All plants were grown under a daily cycle of 16 h light at 22 °C and 8 h of darkness at 16 °C with relative humidity of 75% in a greenhouse and then inoculated at the jointing stage with a mixture of *Bgt* isolates collected from the Wenjiang region (Chengdu, China). When the susceptible control MY11 displayed severe symptoms, the powdery mildew responses of plants were assessed and recorded as infection types (ITs) in a 0–9 scale, where plants with ITs of 0–4 were considered resistant, and those with ITs of 5–9 were susceptible [75].

### 4.6. Agronomic Trait Evaluation

The morphological traits of 18-1-5 and its wheat parents were evaluated in a field trial with three replications at the Wenjiang Experimental Station of Sichuan Agricultural University (Chengdu, China) during the 2021–2022 growing season. For each replicate, 15 grains of each line were evenly planted in 1.5 m rows spaced 0.3 m apart. At the physiological maturity stage, the following yield-related traits were assessed in 15 randomly selected plants of each line: plant height, tiller number, spike length, spikelets per spike, kernels per spike, kernel length, kernel width, and thousand-kernel weight. Significant differences in all measured traits between 18-1-5 and its wheat parents were analyzed using the IBM SPSS Statistics 24.0 software package.

### 4.7. Genotyping-by-Sequencing and Data Analysis

The genomic DNAs of CS*ph2b*, *P. huashanica*, and 18-1-5 were extracted from fresh young leaves using a Hi-DNAsecure DP350 plant kit (Tiangen, Beijing, China) and subjected to GBS (Novogene, Beijing, China). High-quality DNA libraries were constructed and sequenced on an Illumina HiSeq system. The raw reads in FASTQ format were filtered by removing low-quality reads and reads with adapter and/or poly-N sequences to generate clean 140-bp reads. To obtain sequences specific for *P. huashanica* chromosome 7Ns, sequences with less than 18% similarity to CS (IWGSC-RefSeq-v1.0) were first selected. Next, sequences with less than 18% similarity to CS*ph2b* were selected, and finally those with at least 90% similarity to *P. huashanica* were identified as the alien chromosome-specific sequences of 18-1-5.

### 4.8. Development and Verification of PCR-Based Markers

PCR primers for the above-mentioned specific sequences were designed using the online tool Primer3 Plus (accessed on 15 February 2022, http://www.primer3plus.com/cgi-bin/dev/primer3plus.cgi) and then synthesized by Sangon Biotech. After PCR amplification, products were separated on 3% agarose gels. Markers specifically amplified from *P. huashanica* and 18-1-5 but not from CS and CS*ph2b* were regarded as specific for alien chromosomes of 18-1-5. The stability, repeatability, and specificity of these newly developed markers were further validated in 13 wheat-related species with diverse genomes. PCR components were consistent with those described by Tan et al. [25]. The following touchdown PCR protocol was used: 94 °C for 5 min, followed by 10 cycles of 94 °C for 30 s, 65 °C for 30 s (with the temperature decreased by 1 °C per cycle to a final temperature of 60 °C), and 72 °C for 30 s, followed by 28 cycles of 94 °C for 30 s, 60 °C for 30 s, and 72 °C for 30 s, and a final extension at 72 °C for 10 min. 

## 5. Conclusions

In this study, we developed a cytogenetically relatively stable wheat–*P. huashanica* 7Ns disomic addition line from a cross between common wheat and *P. huashanica* and characterized this line by GISH, mc-FISH, ND-FISH, and molecular marker analyses. This line, which confers a high level of resistance to powdery mildew and possesses superior agronomic traits, is a potentially valuable germplasm resource for breeding new disease-resistant wheat cultivars. Moreover, 118 PCR-based markers specific for chromosome 7Ns of *P. huashanica* were successfully developed using the GBS technique. All these markers should be applicable for efficiently tracing *P. huashanica* chromosomes and chromosomal segments during wheat disease-resistant breeding and also for investigating genetic differences and phylogenetic relationships among diverse Ns genomes and other closely related genomes. In future research, we plan to use the CS*ph1b* mutant to generate small segment translocation lines carrying powdery mildew resistance gene(s) without a genetic linkage drag for further breeding applications.

## Figures and Tables

**Figure 1 ijms-23-10285-f001:**
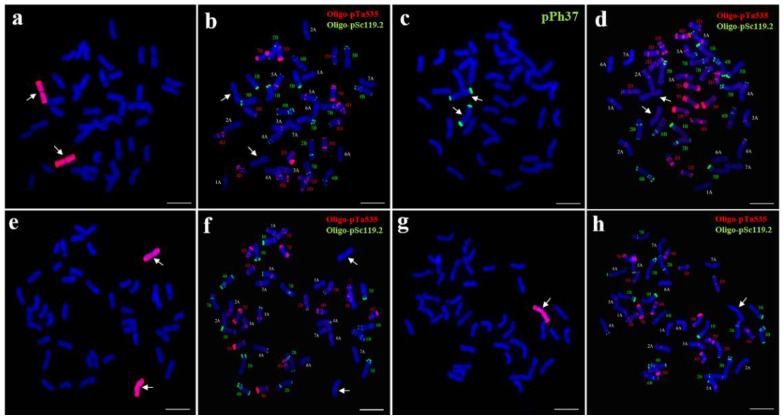
Sequential GISH and mc-FISH analysis of the wheat–*P. huashanica* disomic addition line 18-1-5. (**a**,**e**,**g**) GISH detection of 18-1-5 using *P. huashanica* genomic DNA as a probe (red). (**b**,**d**,**f**,**h**) mc-FISH analysis of 18-1-5 using Oligo-pSc119.2 (green) and Oligo-pTa535 (red). (**c**) FISH identification of 18-1-5 using pPh37 (green). Arrows indicate the introduced *P. huashanica* chromosomes in line 18-1-5. Scale bar: 10 μm.

**Figure 2 ijms-23-10285-f002:**
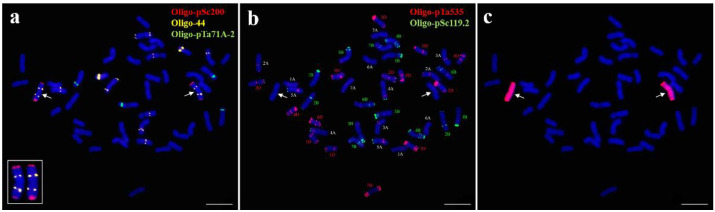
ND-FISH and sequential mc-FISH-GISH analysis of the wheat–*P. huashanica* disomic addition line 18-1-5. (**a**) ND-FISH analysis of 18-1-5 using Oligo-pSc200 (red), Oligo-44 (yellow), and Oligo-pTa71A-2 (green). (**b**) mc-FISH identification on the same metaphase cell of 18-1-5 using Oligo-pSc119.2 (green) and Oligo-pTa535 (red). (**c**) GISH analysis on the same metaphase cell of 18-1-5 using *P. huashanica* genomic DNA as a probe (red). Arrows indicate the introduced *P. huashanica* chromosomes in line 18-1-5. Scale bar: 10 μm.

**Figure 3 ijms-23-10285-f003:**
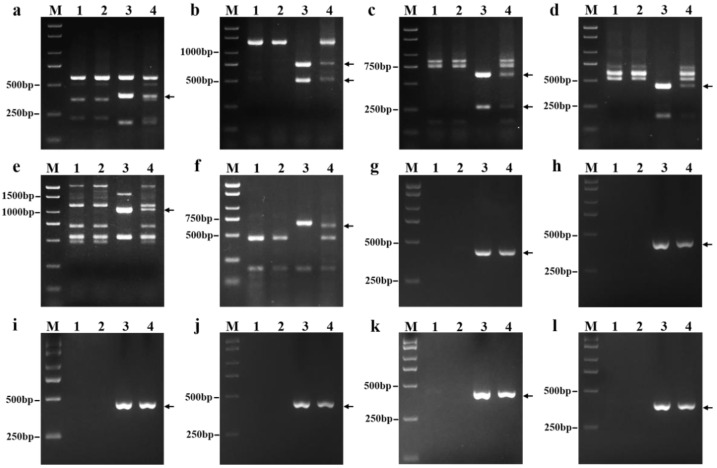
PCR amplification patterns of wheat PLUG and *P. huashanica* 7Ns chromosome-specific markers. (**a**) *TNAC1782-TaqI*; (**b**) *TNAC1803-HaeIII*; (**c**) *TNAC1815-TaqI*; (**d**) *TNAC1867-HaeIII*; (**e**) *TNAC1806-TaqI*; (**f**) *TNAC1845-TaqI*; (**g**) PH7Ns-9; (**h**) PH7Ns-14; (**i**) PH7Ns-38; (**j**) PH7Ns-43; (**k**) PH7Ns-48; and (**l**) PH7Ns-49. (M) Marker (2000 bp); (1) CS; (2) CS*ph2b*; (3) *P. huashanica*; and (4) 18-1-5 (wheat–*P. huashanica* 7Ns disomic addition line). Arrows indicate the diagnostic amplification products for Ns genome.

**Figure 4 ijms-23-10285-f004:**
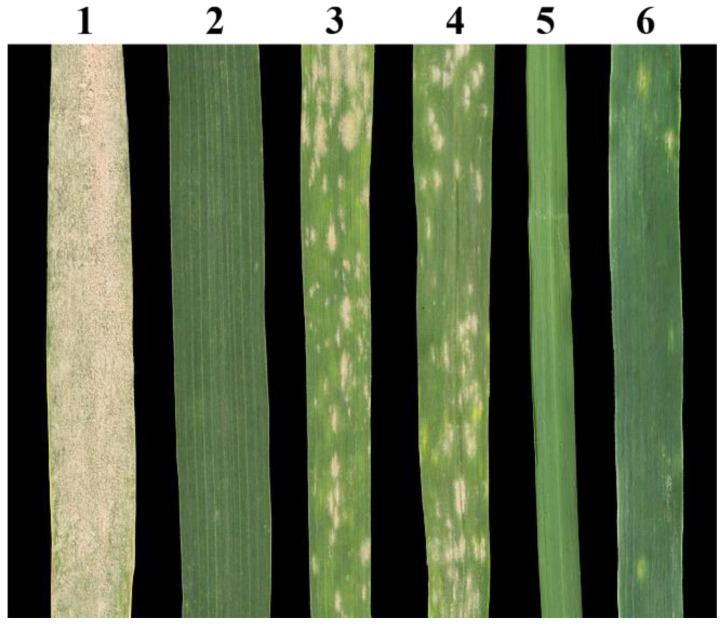
Powdery mildew responses of the wheat–*P. huashanica* disomic addition line 18-1-5, its wheat parents, and the controls at the adult plant stage. (**1**) MY11; (**2**) L658; (**3**) CS; (**4**) CS*ph2b*; (**5**) *P. huashanica*; and (**6**) 18-1-5 (wheat–*P. huashanica* 7Ns disomic addition line).

**Figure 5 ijms-23-10285-f005:**
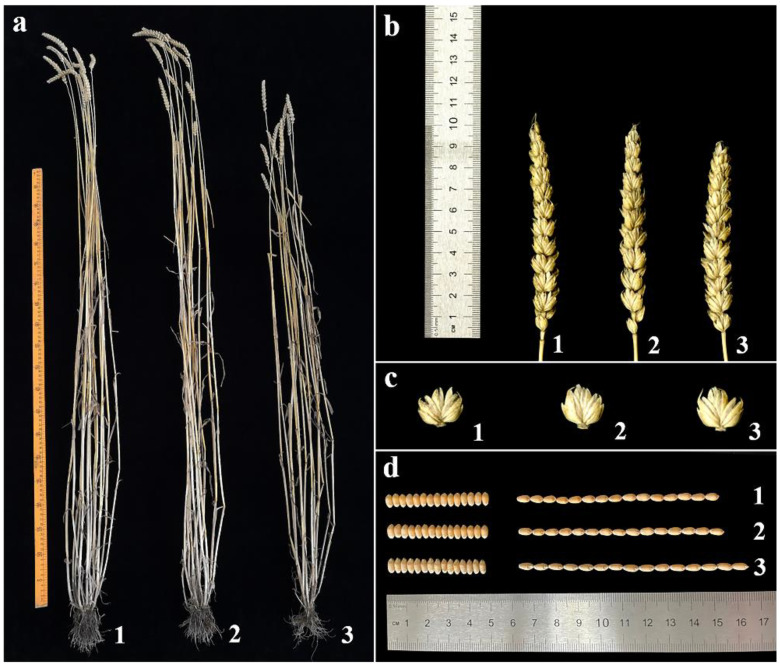
The agronomic traits of the wheat–*P. huashanica* disomic addition line 18-1-5 and its wheat parents. (**a**) Adult plants; (**b**) spikes; (**c**) spikelets; and (**d**) grains. (1) CS; (2) CS*ph2b*; and (3) 18-1-5.

**Figure 6 ijms-23-10285-f006:**
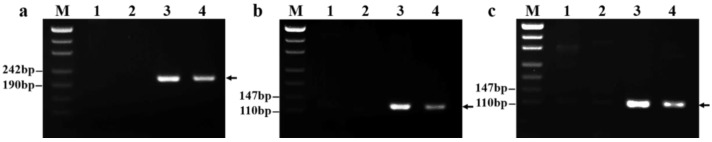
PCR amplification results of *P. huashanica* 7Ns chromosome-specific markers. (**a**) GPH7Ns-155; (**b**) GPH7Ns-300; and (**c**) GPH7Ns-523. (M) Marker (500 bp); (1) CS; (2) CS*ph2b*; (3) *P. huashanica*; (4) 18-1-5 (wheat–*P. huashanica* 7Ns disomic addition line). Arrows show the diagnostic amplification products of *P. huashanica* chromosome 7Ns.

**Figure 7 ijms-23-10285-f007:**
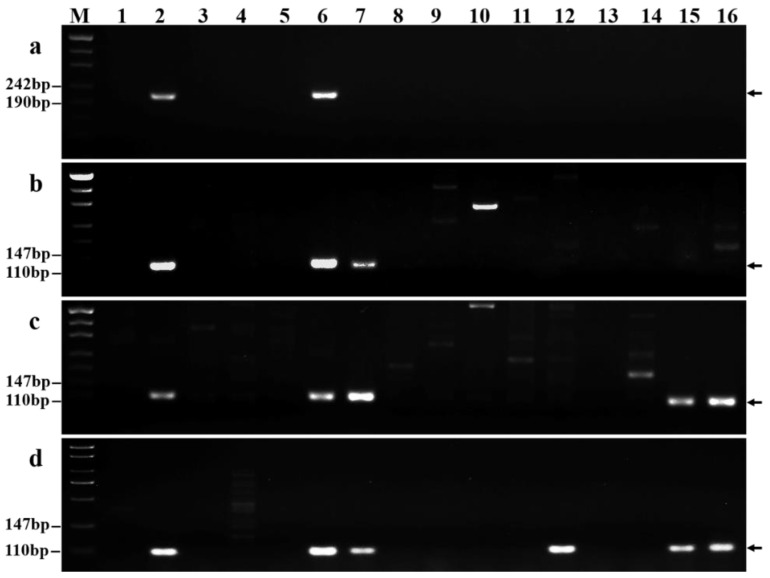
Specificity and stability of *P. huashanica* 7Ns chromosome-specific markers in other wheat-related species. (**a**) GPH7Ns-155; (**b**) GPH7Ns-300; (**c**) GPH7Ns-523; and (**d**) GPH7Ns-350. (M) Marker (500 bp); (1) CS; (2) 18-1-5 (wheat–*P. huashanica* 7Ns disomic addition line); (3) *T. monococcum*; (4) *Ae. speltoides*; (5) *Ae. tauschii*; (6) *P. huashanica*; (7) *P. juncea*; (8) *Th. bessarabicum*; (9) *Th. elongatum*; (10) *D. villosum*; (11) *H. bogdanii*; (12) *Ag. cristatum*; (13) *S. cereale*; (14) *Pse. libanotica*; (15) *L. multicaulis*; and (16) *L. angustus*. Arrows show the diagnostic amplification products of *P. huashanica* chromosome 7Ns.

**Table 1 ijms-23-10285-t001:** The agronomic performance of 18-1-5, and its wheat parents CS and CS*ph2b*.

Lines	Plant Height (cm)	Tiller Number	Spike Length (cm)	Spikelets per Spike	Kernels per Spike	Kernel Length (mm)	Kernel Width (mm)	Thousand-Kernel Weight (g)
CS	140.11 ± 1.28 Aa	8.40 ± 0.51 Aa	8.34 ± 0.13 Aa	21.60 ± 0.31 Aa	53.87 ± 1.32 Aa	5.97 ± 0.03 Aa	3.29 ± 0.02 Aa	30.29 ± 0.26 Aa
CS*ph2b*	133.88 ± 1.63 Bb	7.93 ± 0.37 Aa	8.70 ± 0.13 Aa	22.20 ± 0.30 Aa	55.80 ± 1.68 Aa	6.07 ± 0.05 Aa	3.21 ± 0.03 Ab	29.58 ± 0.27 Aa
18-1-5	111.87 ± 1.74 Cc	7.47 ± 0.58 Aa	8.67 ± 0.15 Aa	20.00 ± 0.37 Bb	46.47 ± 1.59 Bb	6.84 ± 0.08 Bb	2.91 ± 0.03 Bc	27.79 ± 0.62 Bb

Data in the columns indicate means ± standard errors. Different uppercase and lowercase letters following the means indicate significant differences at the *p* < 0.01 and *p* < 0.05 levels, respectively.

**Table 2 ijms-23-10285-t002:** Quality of GBS data.

Sample	Raw Base (bp)	Clean Base (bp)	Effective Rate (%)	Error Rate (%)	Q20 (%)	Q30 (%)	GC Content (%)
CS*ph2b*	1,404,766,080	1,404,693,792	99.99	0.05	94.21	85.77	43.67
ZY3156	1,809,454,176	1,809,426,816	100.00	0.05	94.35	85.76	44.20
18-1-5	2,734,574,112	2,734,299,360	99.99	0.04	96.87	91.07	44.55

**Table 3 ijms-23-10285-t003:** Sequence alignment between 18-1-5 and its parents.

Sample	Total Reads	Unmapped Reads	uniqReads
CS*ph2b*	9,754,818	47,036	
ZY3156	12,565,464	6,476,616	
18-1-5	18,988,190	429,274	15,925

Unmapped reads and uniqReads are those unmapped on Chinese Spring or unique to 18-1-5, respectively.

## Data Availability

The datasets generated and analyzed during the present study are available from the corresponding author on reasonable request.

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
