# Peer review of "Cytogenetic and Molecular Marker Analyses of a Novel Wheat–Psathyrostachys huashanica 7Ns Disomic Addition Line with Powdery Mildew Resistance"

_ijms, 2022, doi:10.3390/ijms231810285_

Round 1
Reviewer 1 Report
Manuscript Title: Cytogenetic and molecular marker analyses of a novel wheat–Psathyrostachys huashanica 7Ns disomic addition line with powdery mildew resistance
The manuscript is very interesting and give the valuable information to the researchers and readers. The subject of the manuscript is consistent with the scope of the Journal. The English language is fluent and easy to read. Thus, I suggested that the manuscript need to be minor revised before it is accepted by this journal.
The following specific comments are observed:
1. The language grammar of the article should be further improved.
2. Compared with the previous researches, what are the advantages of this study using GBS technology to develop molecular markers for Psathyrostachys huashanica?
3. Compared with previous studies, what are the outstanding advantages of the wheat-Psathyrostachys huashanica 7Ns disomic addition line identified in this study?
4. As an intermediate material for breeding, how can the addition line be further used in breeding?
5. In Table 4, ph2b in CSph2b is italic.
Author Response
Response to Reviewer 1 Comments
Dear Reviewer 1:
Thank you for your letter and comments concerning our manuscript entitled ‘Cytogenetic and molecular marker analyses of a novel wheat–Psathyrostachys huashanica 7Ns disomic addition line with powdery mildew resistance’ (ID: ijms-1869340). Those comments are all valuable and very helpful for revising and improving our paper, as well as the important guiding significance to our researches. We have studied comments carefully and have made correction which we hope meet with approval. Revised portion are marked up using the “Track Changes” function in the manuscript. The main corrections in the manuscript and the responds to your comments are as follows.
Sincerely yours,
Houyang Kang
Point 1: The manuscript is very interesting and give the valuable information to the researchers and readers. The subject of the manuscript is consistent with the scope of the Journal. The English language is fluent and easy to read. Thus, I suggested that the manuscript need to be minor revised before it is accepted by this journal.
Response 1: Thank you very much for your high praise of our manuscript and your valuable comments.
Point 2: The language grammar of the article should be further improved.
Response 2: Thank you for your valuable suggestion. We have professionally touched up the language grammar of the manuscript, and hope it can meet your requirements.
Point 3: Compared with the previous researches, what are the advantages of this study using GBS technology to develop molecular markers for Psathyrostachys huashanica ?
Response 3: Many thanks for your comments. In the previous researches, Psathyrostachys huashanica genome- or chromosome-specific molecular markers developed by conventional methodologies, including random-amplified polymorphic DNA and sequence-characterized amplified region techniques, are inefficient, inaccurate, and costly. In addition, the number of existing markers specific to Psathyrostachys huashanica has been totally insufficient to satisfy research demands. The GBS approach, a high-throughput, high accuracy, low cost, and next-generation sequencing-based technology, has been applied successfully to develop large numbers of highly accurate molecular markers in Triticeae species without reference genome sequences, such as for tetraploid Thinopyrum elongatum chromosomes 1E and 4E (Li et al. 2019; Gong et al. 2022). In the meantime, the molecular markers developed by GBS technique are also more stable and specific, and can be used for efficient marker-assisted breeding.
Point 4: Compared with previous studies, what are the outstanding advantages of the wheat-Psathyrostachys huashanica 7Ns disomic addition line identified in this study ?
Response 4: Many thanks for your comments. The wheat-Psathyrostachys huashanica 7Ns disomic addition line 18-1-5 identified in this study is highly resistant to powdery mildew and possesses superior agronomic traits, and this is the new report of a successful transfer of a high-level powdery mildew resistance gene from chromosome 7Ns of P. huashanica. Therefore, the line is a potentially valuable germplasm resource for the genetic improvement of wheat powdery mildew resistance.
Point 5: As an intermediate material for breeding, how can the addition line be further used in breeding?
Response 5: Many thanks for your comments. The addition line 18-1-5, an important intermediate material for breeding, cannot be directly used in wheat production due to unfavourable linkage drag. In contrast, translocation lines are more suitable for practical breeding on account of the smaller amount of alien genetic material, less linkage drag, and regular meiotic behavior. In future research, we plan to use the CSph1b mutant to generate small segment translocation lines carrying powdery mildew resistance gene(s) without a genetic linkage drag for further breeding applications.
Point 6: In Table 4, ph2b in CSph2b is italic.
Response 6: Thank you for your careful work. We are very sorry for the errors, and have revised it in Table 4.
Reviewer 2 Report
In this study, the authors generated a cross between common wheat and P. huashanica that produced a wheat-P.huashanica 7Ns disomic addition line, which was then characterized by GISH, mc-FISH, ND-FISH, and molecular marker studies. This line is a potentially significant genetic resource for developing novel disease-resistant wheat cultivars, and the obtained results are useful to the wheat research community.
· To make the result more exactly, the author needs to add the PCR amplification results of reference gene (such as the Actin gene) as a control to verify the wheat DNA and PCR amplification system and program since the result of no band could be generated by many other factors, besides to the allele variation.
· Table 1 should be moved as supplemental information
· Table 2; Need proper alignment and explain the letter Aa and Bb,
· Add the flowering and single plant yield data in Table 2
· There are some grammar issues that should be fixed in order to aid the accessibility of the review to the reader.
Author Response
Response to Reviewer 2 Comments
Dear Reviewer 2:
Thank you for your letter and comments concerning our manuscript entitled ‘Cytogenetic and molecular marker analyses of a novel wheat–Psathyrostachys huashanica 7Ns disomic addition line with powdery mildew resistance’ (ID: ijms-1869340). Those comments are all valuable and very helpful for revising and improving our paper, as well as the important guiding significance to our researches. We have studied comments carefully and have made correction which we hope meet with approval. Revised portion are marked up using the “Track Changes” function in the manuscript. The main corrections in the manuscript and the responds to your comments are as follows.
Sincerely yours,
Houyang Kang
Point 1: In this study, the authors generated a cross between common wheat and P. huashanica that produced a wheat-P.huashanica 7Ns disomic addition line, which was then characterized by GISH, mc-FISH, ND-FISH, and molecular marker studies. This line is a potentially significant genetic resource for developing novel disease-resistant wheat cultivars, and the obtained results are useful to the wheat research community.
Response 1: Many thanks for your comments. We agree with you that the addition line is a potentially significant genetic resource for developing novel disease-resistant wheat cultivars. In the future, further work will be done to create more small fragment translocation lines carrying powdery mildew resistance gene(s) without a genetic linkage drag for further breeding applications.
Point 2: To make the result more exactly, the author needs to add the PCR amplification results of reference gene (such as the Actin gene) as a control to verify the wheat DNA and PCR amplification system and program since the result of no band could be generated by many other factors, besides to the allele variation.
Response 2: Thank you for your valuable and thoughtful suggestion. In this study, the negative controls CS and CSph2b, and positive control P. huashanica have been used in all PCR amplification analyses, and we repeatedly confirmed that our results are correct and accurate. Therefore, there is no need to add the PCR amplification results of reference gene as a control.
Point 3: Table 1 should be moved as supplemental information
Response 3: According to your suggestion, we removed Table 1 in the manuscript, and changed it as a new supplementary information (Table S1). Due to the addition of Table S1 and the deletion of Table 1, the original Table S1 to Table S2 and Table 2 to Table 4 were re-numbered as Table S2 to Table S3 and Table 1 to Table 3, respectively, and revised in the manuscript.
Point 4: Table 2; Need proper alignment and explain the letter Aa and Bb,
Response 4: Thank you for your careful work. We did the proper alignment for Table 2. The meaning of the letter Aa and Bb have been explained in note to Table 2.
Point 5: Add the flowering and single plant yield data in Table 2
Response 5: Many thanks for your comments. In the present study, eight yield-related traits, including plant height, tiller number, spike length, spikelets per spike, kernels per spike, kernel length, kernel width, and thousand-kernel weight, were assessed for 15 randomly selected plants of each line. Of these, three yield factors, comprising tiller number, kernels per spike, and thousand-kernel weight, could comprehensively reflect yield per plant. Most importantly, the addition line, which is an intermediate material for breeding, cannot be directly used in wheat production due to unfavourable linkage drag. Accordingly, we only measured the above-mentioned eight main traits except the flowering stage and single plant yield for assessing the effect of the introduction of alien 7Ns chromosomes on mian agronomic traits of common wheat.
Point 6: There are some grammar issues that should be fixed in order to aid the accessibility of the review to the reader.
Response 6: Thank you for your valuable suggestion. We revised some grammar errors in the manuscript, and hope it will get your approval.
Reviewer 3 Report
This manuscript entitled 'Cytogenetic and molecular marker analyses of a novel wheat–Psathyrostachys huashanica 7Ns disomic addition line with powdery mildew resistance' is well written and brings very good value to our research. The outcomes of the study can be of great value to tackle Pm disease. I only suggest the author to make one table illustrating the traits that Psathyrostachys huashanica is donor for.
Author Response
Response to Reviewer 3 Comments
Dear Reviewer 3:
Thank you for your letter and comments concerning our manuscript entitled ‘Cytogenetic and molecular marker analyses of a novel wheat–Psathyrostachys huashanica 7Ns disomic addition line with powdery mildew resistance’ (ID: ijms-1869340). Those comments are all valuable and very helpful for revising and improving our paper, as well as the important guiding significance to our researches. We have studied comments carefully and have made correction which we hope meet with approval. Revised portion are marked up using the “Track Changes” function in the manuscript. The main corrections in the manuscript and the responds to your comments are as follows.
Sincerely yours,
Houyang Kang
Point 1: This manuscript entitled 'Cytogenetic and molecular marker analyses of a novel wheat–Psathyrostachys huashanica 7Ns disomic addition line with powdery mildew resistance' is well written and brings very good value to our research. The outcomes of the study can be of great value to tackle Pm disease. I only suggest the author to make one table illustrating the traits that Psathyrostachys huashanica is donor for.
Response 1: Many thanks for your high praise of our manuscript. We agree with you that the outcomes of the study can be of great value to tackle Pm disease in wheat disease-resistant breeding. We have illustrated the traits of Psathyrostachys huashanica in the ‘Introduction’ section in the manuscript, such as resistance to multiple wheat diseases (scab, stripe rust, leaf rust, powdery mildew, and take-all), tolerance to abiotic stress (drought, salinity, and alkalinity), early maturation, and dwarf stature. Additionally, as an important gene donor for wheat genetic improvement, all seven Ns chromosomes from P. huashanica that have been introduced into common wheat exhibit excellent characteristics with regard to disease resistance and agronomic traits, which were also illustrated in the ‘Discussion’ section.
Round 2
Reviewer 2 Report
The authors have revised the manuscript according to the Reviewer's comments. I have no other comment for this manuscript.